# The 2021 Greece Central Crete $M_L$ 5.8 Earthquake: An Example of Coalescent Fault Segments Reconstructed from InSAR and GNSS Data

Nicola Angelo Famiglietti [1], Zeinab Golshadi [2], Filippos Vallianatos [3,4], Riccardo Caputo [4,5], Maria Kouli [4], Vassilis Sakkas [3], Simone Atzori [6], Raffaele Moschillo [1], Gianpaolo Cecere [1], Ciriaco D'Ambrosio [1] and Annamaria Vicari [1,*]

1 Istituto Nazionale di Geofisica e Vulcanologia, Sezione Irpinia, 83035 Grottaminarda, Italy
2 Institute of Geophysics, University of Tehran, Tehran 141556466, Iran
3 Section of Geophysics–Geothermics, Department of Geology and Geoenvironment, National and Kapodistrian University of Athens, 15784 Athens, Greece
4 Institute of Physics of the Earth's Interior and Geohazards, UNESCO Chair on Solid Earth Physics and Geohazards Risk Reduction, Hellenic Mediterranean University Research Center, Crete, 73133 Chania, Greece
5 Dipartimento di Fisica e Scienze Della Terra, Università Degli Studi di Ferrara, 44121 Ferrara, Italy
6 Istituto Nazionale di Geofisica e Vulcanologia, Osservatorio Nazionale Terremoti, 00143 Rome, Italy
* Correspondence: annamaria.vicari@ingv.it; Tel.: +39-0825446057

**Abstract:** The $M_L$ 5.8 earthquake that hit the island of Crete on 27 September 2021 is analysed with InSAR (Interferometry from Synthetic Aperture Radar) and GNSS (Global Navigation Satellite System) data. The purpose of this work is to create a model with sufficient detail for the geophysical processes that take place in several kilometres below the earth's surface and improve our ability to observe active tectonic processes using geodetic and seismic data. InSAR coseismic displacements maps show negative values along the LOS of ~18 cm for the ascending orbit and ~20 cm for the descending one. Similarly, the GNSS data of three permanent stations were used in PPK (Post Processing Kinematic) mode to (i) estimate the coseismic shifts, highlighting the same range of values as the InSAR, (ii) model the deformation of the ground associated with the main shock, and (iii) validate InSAR results by combining GNSS and InSAR data. This allowed us to constrain the geometric characteristics of the seismogenic fault and the slip distribution on it. Our model, which stands on a joint inversion of the InSAR and GNSS data, highlights a major rupture surface striking 214°, dipping 50° NW and extending at depth from 2.5 km down to 12 km. The kinematics is almost dip-slip normal (rake −106°), while a maximum slip of ~1.0 m occurred at a depth of ca. 6 km. The crucial though indirect role of inherited tectonic structures affecting the seismogenic crustal volume is also discussed suggesting their influence on the surrounding stress field and their capacity to dynamically merge distinct fault segments.

**Keywords:** SAR; GNSS; interferometry; source modelling; 27 September 2021 earthquake; Crete system faults

## 1. Introduction

The location of Crete Island falls within one of the most important seismically active areas in the world, north of the Hellenic Arc (Figure 1a). This major geodynamic feature formed as a consequence of the Nubia–Eurasia convergence [1–3]. Nowadays, the central sector of the Hellenic Arc, south of Crete, is represented by the so-called East Mediterranean Ridge [4,5]. As a result of the rapid S-W movement of the southern Aegean with respect to Eurasia, the Mediterranean oceanic crust subducts northwards with a velocity of 35 mm/a (which greatly exceeds the convergence rate between Africa and Eurasia, approximately 5–10 mm/a) below Crete and the Peloponnese [6]. The tectonics of Crete Island is currently

dominated by crustal extension likely as a result of the slab retreat [7], the Aegean mantle wedge intrusion [8] and the subsequent strong uplift [9].

Stretching directions are, however, not regionally uniform [10] as clearly documented by the variable trend of the major active normal faults affecting the island of Crete [11] and its surroundings [12]. On 27 September 2021, a moderate $M_L$ 5.8 earthquake affected Central Crete (Greece) not far from the city of Heraklion (Figure 1). The structural damage of the villages close to the epicentral area, located near Arkalochori, was considerable, and there were several injured and one person died; furthermore, the shaking effects of the mainshock have been widely felt across the island. Within the first 24 h after the mainshock, an $M_L$ 5.2 earthquake and several $M_L$ 4+ aftershocks occurred (Table 1).

**Table 1.** Mainshock and aftershock (M > 4.5) focal mechanism of the 2021 Central Crete seismic sequence (National Observatory of Athens—NOA data) immediately following the earthquake.

| Magnitude ($M_L$) & Focal Mechanisms (NOA) | Date & Time (UTC) | Location | Latitude (°N) | Longitude (°E) | Depth (km) |
|---|---|---|---|---|---|
| 5.8 (main) | 2021/09/27 06:17:21 | 23.3 km SE of Heraklion | 35.1512 | 25.2736 | 10 |
| 4.5 | 2021/09/27 07:30:45 | 23.7 km SSE of Heraklion | 35.1334 | 25.2457 | 14 |
| 4.7 | 2021/09/27 11:02:25 | 19.5 km SE of Heraklion | 35.1805 | 25.2525 | 13 |
| 5.2 | 2021/09/28 04:48:09 | 20.8 km SSE of Heraklion | 35.1540 | 25.2232 | 11 |
| 4.6 | 2021/09/28 15:13:15 | 23.4 km SSE of Heraklion | 35.1466 | 25.2663 | 14 |
| 4.6 | 2021/09/29 11:54:49 | 21.3 km SSE of Heraklion | 25.2058 | 25.1561 | 16 |

The crustal volume that was reactivated by the 2021 seismic sequence is at present stretched along a ca. ESE-WNW direction as it is well documented by recent geodetic investigations, based on both GNSS and InSAR data [13–15] and by the seismological information about the sequence [16,17] confirming a mainly dip-slip normal kinematics.

The distribution of the epicentres partially overlaps the southwestern sector of the Kastelli seismogenic source included in the Greek Database of Seismogenic Sources (GRCS743) [12] for which has been estimated a maximum magnitude of 6.4. The fault trace of this tectonic structure has been mapped for about 22 km showing a curved shape geometry with ENE-WSW strike in the NE sector and NNE-SSW orientation towards its SW termination. Based on morphotectonic analysis and empirical relationships [16], the estimated mean recurrence interval is about 812 years over the last 13 ka with maximum vertical displacements of 65–70 cm. Among the several faults affecting the eastern sector of the Heraklion Basin with variable settings, the Kastelli Fault has been considered the most active tectonic structure [17] and this is likely confirmed by microseismic activity recorded in the area [18].

It should be noted that different names have been attributed to the faults bordering to the east the Heraklion Basin generating some confusion in the recent literature. Indeed, the name "Kastelli Fault" (as indicated in Figure 1) was firstly proposed in 2001 by [19] referring to "one of the most impressive structures of the Heraklion basin", while in 2006 [11] the same tectonic structure was mapped in some detail and clearly characterised from structural, morphotectonic and seismotectonic view points. Following also [18], the most popular databases of seismogenic [12] and active [20] faults use the same nomenclature,

which has been also largely applied in the international literature (e.g., [17,19,21–24]. More recently, and likely following an unpublished work by [25] where however the fault is never cartographically represented, other authors refer to the Kastelli Fault (as it is plotted in Figure 1c), as the Geraki Fault (or Fault Zone) (e.g., [13,15,16]), while a minor tectonic structure with poor evidence of recent reactivations, running west of the major morphotectonic escarpment and its post-Last Glacial Maximum (LGM) free face, is referred to as 'Kastelli Fault'. In agreement with [20], this secondary structure is instead referred to as Agni Fault. Similarly, its southern segment is called Avli Fault (Figure 1c), though it is sometimes indicated as Lagoutas Fault [13]. In this paper, the used faults terminology is according to the structures shown in Figure 1c.

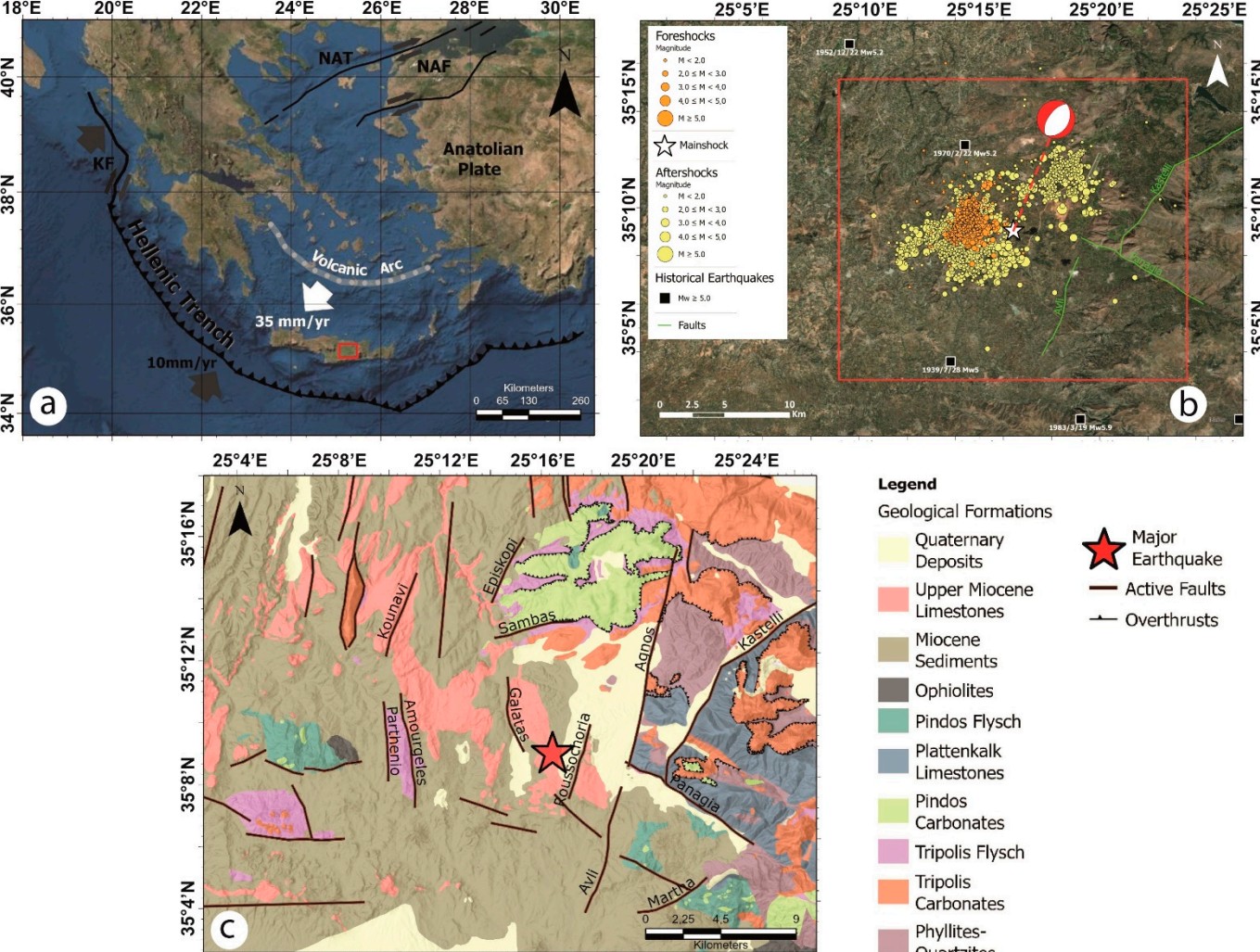

**Figure 1.** (**a**) Indicative map of the main structural characteristics of the Hellenic Arc and Trench system. KF: Kefalonia Transform Fault, NAF: North Anatolian Fault, NAT: North Anatolian Trench. The white thick arrow shows the direction of relative motion between the Aegean and Nubian plates accommodated by the Hellenic subduction (Reprinted/adapted with permission from Ref. [20]). The red box indicates the study area. (**b**) Map of the investigated area showing the epicenter (white star) and the focal mechanisms of the mainshock, the M > 5 historical earthquakes from 1900 up to today (https://www.gein.noa.gr/en/services-products/earthquake-catalogs/ (accessed on 23 September 2022)), the foreshocks and the aftershocks (from 1 June to 18 October 2021 relocated by [13]) and the composite seismogenic sources included in GreDaSS [12]; (**c**) Simplified geological map of the broader area [21] draped on a shaded relief, with the traces of the major active faults [22] affecting the area.

Several reporting agencies provided moment tensor solutions for the mainshock (visit of 1 October 2021 to the portal https://www.seismicportal.eu/mtws/), suggesting that the activated normal fault had a mean NW dip-direction with a dip angle of about 54°, in agreement with the geological observations [11,13,16,19]. Nearly four months prior to the occurrence of the mainshock, several foreshocks had been recorded in the broad area [26], and several aftershocks occurred in the days and months that followed. From 1 June to 24 July, about 155 foreshocks were recorded with magnitudes up to $M_L$ 4.3, with four exceeding $M_L$ 4.0. At 02:07:37 UTC on 24 July, a moderate pre-shock with magnitude $M_L$ 4.8 and hundreds of foreshocks with magnitudes up to 3.8 occurred until the main event. In the following days, several major aftershocks occurred, about eight of magnitude greater than or equal to $M_L$ 4.2, of which the largest occurred on 28 September at 4:48:09 UTC ($M_L$ 5.3). Consequently, thousands of people suffered damages as a result of the continuous seismic activity. The most serious damage occurred near the village Arkalochori.

In the present research, the main seismic event with $M_L$ 5.8 was analyzed using ground displacement data derived from InSAR (Interferometry from Synthetic Aperture Radar) [23] and GNSS (Global Navigation Satellite System) [24] techniques in order to constrain the fault kinematic and calculate the slip distribution of the rupture surface following the mainshock.

The causative fault parameters of the earthquake were determined by a nonlinear inversion of InSAR and GNSS displacement data, and the slip distribution of the source was determined by using a linear inversion algorithm.

Beyond the contribution to our seismotectonic knowledge, a second major goal of the present work is to investigate the compatibility of the causative fault with previously known local structures and/or the identification of new potential structures.

## 2. Materials and Methods

The data analysis of the 2021 Central Crete sequence is based on the following three steps:

- a two-pass SAR phase interferometric analysis to get the surface displacement field;
- determination of punctual 3D coseismic offset through the differential analysis of GNSS data;
- reconstruction of the source model through joint inversion of InSAR and GNSS data.

### 2.1. InSAR Analysis

The InSAR analysis is based on SAR images of Sentinel-1 satellites, ESA (European Space Agency) constellation, in IW (Interferometric Wide) mode and V-V polarisation.

Four images, in pairs of two and with a temporal distance of 12 days, along the ascending and descending orbits (Table 2), were used to retrieve the coseismic displacement field. Both pairs include 3 days of aftershocks, expected to give a negligible contribution compared to the mainshock.

**Table 2.** Sentinel-1 images used for InSAR processing of $M_L$ 5.8 Central Crete Earthquake.

| Event | Interferogram Number | Pre-Event Date | Post-Event Date | Orbit | Incidence Angle | Track |
|---|---|---|---|---|---|---|
| $M_L$ 5.8–23.3 km SE of Heraklion | 1 | 18/09/2021 | 30/09/2021 | ascending | 3,555,742 | 29 |
| 2021-09-27 06:17:21 (UTC) | 2 | 18/09/2021 | 30/09/2021 | descending | 38,352,143 | 109 |

For each pair, images have been co-registered and averaged 4 times in the satellite range direction, to increase the signal-to-noise ratio and to obtain a final resolution of about 30 m in the geocoded geometry; then orbital corrections were applied using ESA PO (Precise Orbits) and the SRTM-1 digital elevation model was used to remove the topographic phase contribution.

The application of the adaptive filtering algorithm to the raw interferograms allowed us to significantly increase the quality of interferometric fringes, by reducing the phase

noise [25]. Through the MCF (Minimum Cost Flow) algorithm the unwrapping interferogram was obtained [27]. The unwrapped maps were then geocoded with the same SRTM-1 DEM with a pixel resolution of 30 m.

### 2.2. GNSS Analysis

In the current study, the GNSS data were used both to estimate the co-seismic shifts, and to model the ground deformation associated with the mainshock by combining GNSS and InSAR data for analysis, in addition to validation of the InSAR results [28–34].

We processed the data of three permanent GNSS stations on the island of Crete, belonging to the commercial network of METRICA SA (HexagonSmartNet), in PPK (Post Processing Kinematic) mode using the RTKLib software [35] (Figure 2):

- ARKL located in the epicentral area (used as a Rover);
- HERA and MOI1 (located at Heraklion and Mires, respectively), which are both outside the area that suffered instability following the earthquake (used as Bases).

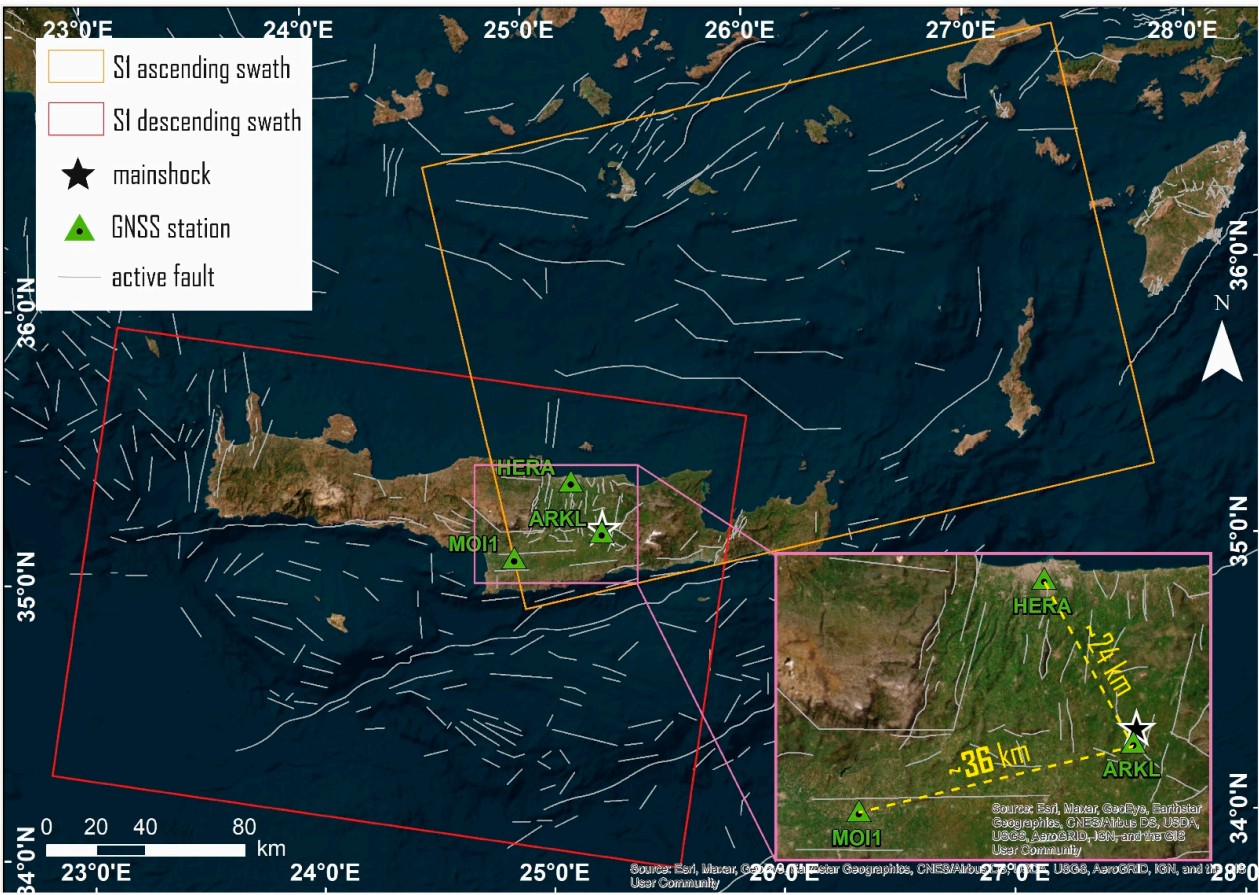

**Figure 2.** The location of the three permanent GNSS Crete Stations and IW Swath of Sentinel 1 images. Basemap reprinted/adapted with permission from Sources: Esri, DigitalGlobe, GeoEye, i-cubed, USDA FSA, USGS, AEX, Getmapping, Aerogrid, IGN, IGP, swisstopo, and the GIS User Com-munity.

The HERA and MOI1 stations, used as reference bases in this configuration, have a distance from the Rover station (ARKL) approximately 24 and 36 km, respectively.

The characteristics of the equipment of the three GNSS stations are:

- HERA: Receiver LEICA GRX1200 + GNSS, Antenna LEIAR10 NONE;
- ARKL: Receiver LEICA GR10, Antenna LEIAR10 NONE;
- MOI1: Receiver LEICA GR30, Antenna LEIAR10 NONE.

The sampling interval was 30 s, and the data include about 12 h of observations. For PPK GNSS processing the navigation engine of RTKLIB has been used, which is based on

the extended Kalman filter (EKF) and double differencing with respect to a nearby base station. In this way, it is possible to reach accuracies in the positioning at the centimetre level, if not too long baselines are used and the ambiguities of phase integers of the carrier are correctly resolved [36]. However, it has been shown that accuracies of a few centimetres can be reached even with greater distances (>25 km) as long as the correct parameters are properly set [37].

As described below, the processing of GNSS data with the PPK method allowed an in-depth analysis of the kinematics of the ARKL station with respect to two different bases and highlighted the usefulness of this method in evaluating the earthquake effects on the ground.

### 2.3. Source Modelling Analysis

The fault modelling technique builds on the joint inversion scheme and developed to create single fault earthquake source representations using geodetic data. The geodetic data are based on SAR images of Sentinel-1 satellites and GNSS data. The geodetic data were inverted for a single double-couple source with nine source parameters; fault strike, dip, rake, average slip, length, width, centroid longitude, latitude and depth. The process of processing both data sets is described below: data modeling was conducted over a set of points regularly sampled from the ascending and descending raster displacement maps with two different spatial resolutions: 500 m in the fault near-field and 2000 m in the far field.

InSAR LOS co-seismic points were then jointly modelled with 3D GNSS points using a dual step validated approach: at first a non-linear inversion was carried out to trace the geometry and position of the fault, using a uniform dislocation value; then we applied a linear inversion to calculate the slip distribution on the inverted fault plane, subdivided into square elements and opportunely extended to include the whole distribution, from the peak value to zero. In both inversions, the underlying geophysical model was used to predict the surface displacement is the elastic dislocation induced by a finite source in a homogeneous half-space [38], with the Williams and Wadge [39] approach to account for the local topography.

The linear inversion is conducted with the additional Non-Negative Least-Square (NNLS) constraint, preventing unrealistic back-slip values, and the inclusion of a regularisation contribution, opportunely weighted with a trial-and-error damping factor [40–43]. Details about both non-linear and linear inversion algorithm implementations can be found in [44,45].

An iterative procedure of the Levenberg–Marquardt optimization algorithm was used for non-linear inversion [46]. Based on the parameter ranges given, the optimization uses the weighted squares of the residuals to minimize the objective function *F*:

$$F = \frac{1}{N} \sum_{i=1}^{N} = \frac{1}{N} \sum_{i=1}^{N} [(d_i - f(m))/\sigma_i]^2 \tag{1}$$

where $d_i$ is the observation value of the *i*-th data point; $\sigma_i$ represents the standard deviation relating to each datum; *m* represents the model parameters vector and *f* is the non-linear forward Okada's model [38] in the inversion that consists of *N* points. As part of this approach, the model parameter vector m is defined in order to minimize *F*. The cost function is a weighted mean of the residuals between observed and predicted data sets. Using multiple restarts, the minimization algorithm can reasonably guarantee catching the global minimum.

As part of the linear inversion procedure, we maintained the fixed geometric settings derived from our preferred non-linear inversion to get the slip distribution along the fault. During this procedure, the fault spread out until the slip vanished to zero and it was subdivided into small patches. Each patch's slip value was obtained from joint inversion of the ascending and descending InSAR and GNSS datasets [44]. We used a trial-and-

error approach for system damping to avoid backslip, in which the empirical parameter is balancing the slip distribution roughness and the data fit [41,42].

## 3. Results

### 3.1. InSAR Results

The dates relating to the used SAR acquisitions have a temporal baseline of 12 days [47], therefore including six events (M > 4.5) of the seismic sequence (in particular, the mainshock of 2021-09-27 at 06:17:21 (UTC) and five aftershocks that took place in the following hours and days) (Tables 1 and 2).

Thanks to this short time interval, as shown in (Figure 3), the produced interferograms have good coherence (>0.6) gaps and prove to be similar to each other. In the production of differential interferograms, the coherence factor is strongly influenced by the temporal baseline and the spatial decorrelation between the reference image and the repeated images. Considering the 12-days time interval and the good quality of Sentinel images, the high level of coherence (>0.6) was maintained across the whole investigated area [48].

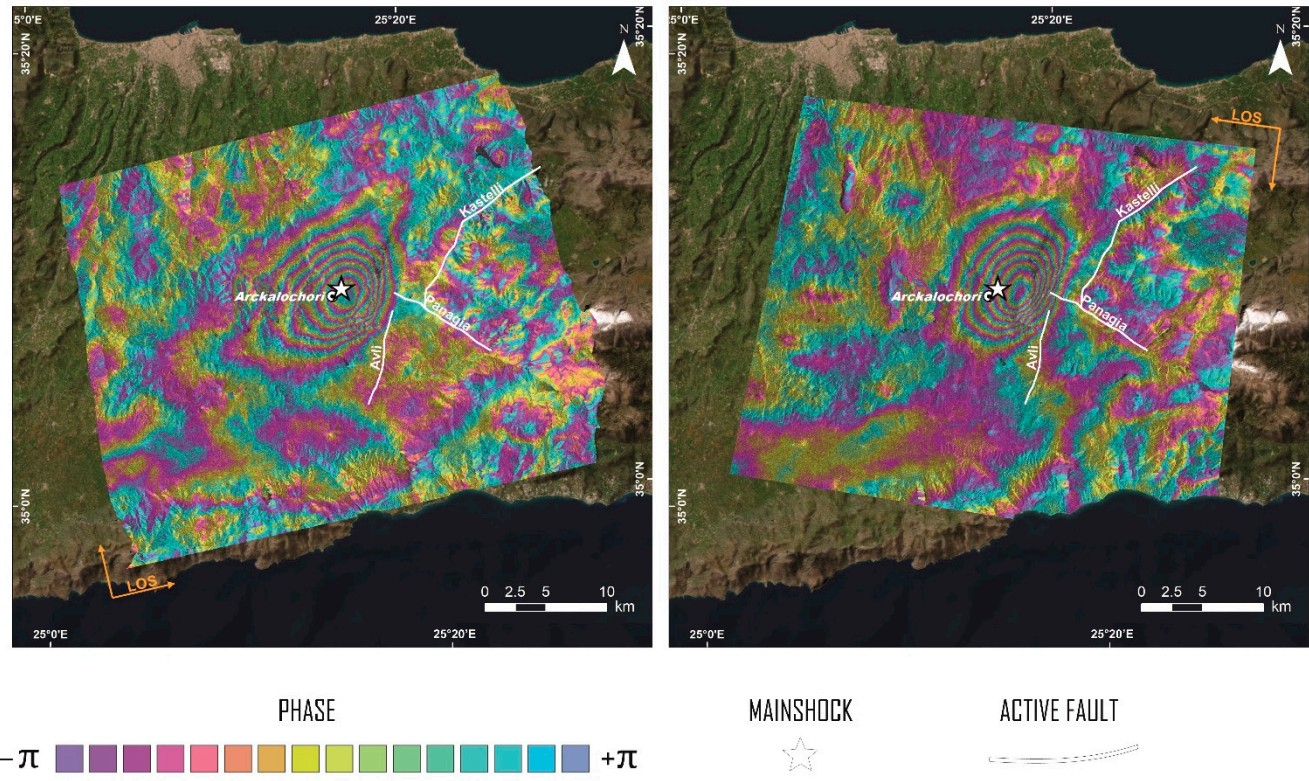

**Figure 3.** Ascending (**left**) and descending (**right**) wrapped interferograms of the 2021 M 5.8 Heraklion earthquake.

The displacement maps along both orbits show a very similar deformation pattern, indicating that the actual ground movement is predominantly vertical, as also visible from the conversion from ascending and descending to vertical and horizontal components [49,50].

More intense displacements occurred at the epicentral area of the mainshock where there is a lowering of the ground up to 20 cm (Figure 4).

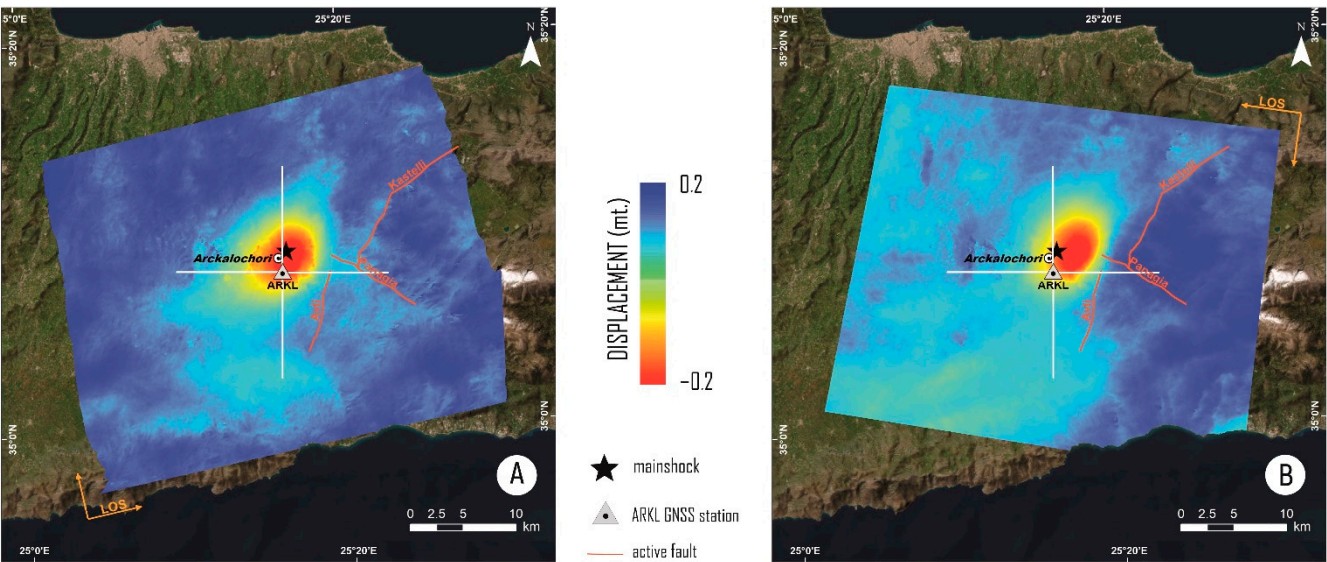

**Figure 4.** Ascending (**A**) and descending (**B**) displacements. The mainshock is highlighted with a black star.

These maps show a peak of displacement (red colour) moving away along LOS direction in the vicinity of the city of Arkalochori, explaining the extensive structural damages in the neighboring villages. Four maps were produced relative to the two transects (Figure 4), two for each orbital direction. The transects were traced in the N–S (North–South) and E–W (East–West) directions, and their crossing point coincides with the position of the ARKL GNSS station. The elevation profiles of the two orbital directions, ascending and descending, were obtained from the two transects. Profiles on North–South and East–West directions were produced to examine the displacement field along these directions. In each profile, the results obtained with the two softwares (SARscape and SNAP, represented by red and blue curves, respectively, in Figure 5) were compared [51]. In particular, negative displacements along LOS (away from the sensor) of ~18 cm for the ascending orbit and ~20 cm for the descending one are highlighted. Moreover, it has to be emphasised the good agreement of the epicentral area of the $M_L$ 5.8 event with the maximum displacement area depicted in all profiles.

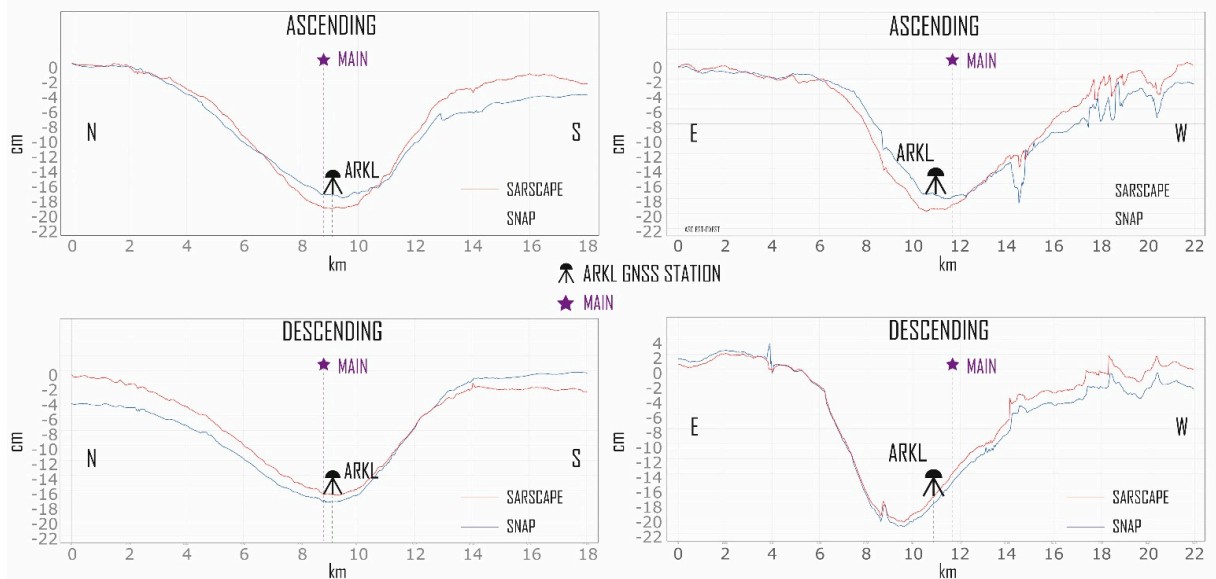

**Figure 5.** Cross sections along the S–N and E–W transects shown in Figure 4.

## 3.2. GNSS Results

The GNSS data were analysed using a differential kinematic post-processing (PPK) approach with the RTKPOST ver.2.4.3 b34 GNSS Post-Processing Software. Two baselines were then calculated:

- ARKL-HERA; with ARKL acting as rover and HERA being the base;
- ARKL_MOI1; with ARKL also being the rover and MOI1 set as base.

Since the data from the ARKL station show an interruption during the seismic event (from 06:17:30 to 06:30:30), for each of the two baselines, about 6 h pre- and post-event were processed: that is to say, from 00:00 to 06:17 and from 06:30 to 12:00 of 27/09/2021, as it can be seen in Figure 6. The GNSS data (of GPS and GLONASS constellations only) were processed using IGS precise orbits [52] in Kinematic mode with an automatically combined of forward and backward directions. This procedure maximises the accuracy of the solutions and improves the quality control [53]. Based on this processing strategy the following results were obtained: ARKL-HERA baseline phase ambiguities fix at 99.3%, while ARKL-MOI1 92.8% fix. From the linearly fitted positions of the pre- vs post-solution, we estimated the three components (E–W, N–S and U–D) of the displacement vectors using the Vincenty formula [54,55].

The GNSS data act not only as GCPs (Ground Control Points) during the processing of the phase SAR data, but also validate the InSAR products [56]. The displacements deriving from the interferograms were thus compared with the co-seismic offsets obtained from the geodetic data (vertical and E–W components), as represented in Figure 7.

The figure shows the agreement between the InSAR and GNSS data along the U–D and E–W directions: the largest subsidence value is highlighted in the epicentral area and is equal to about 20 cm, a more significant displacement of 11 cm towards the east of the footwall block and about 7 cm towards the west of the hanging-wall block confirming an important E–W component of crustal stretching. The co-seismic offsets obtained from GNSS data were projected into LOS (ascending and descending) direction and are reported in Table 3.

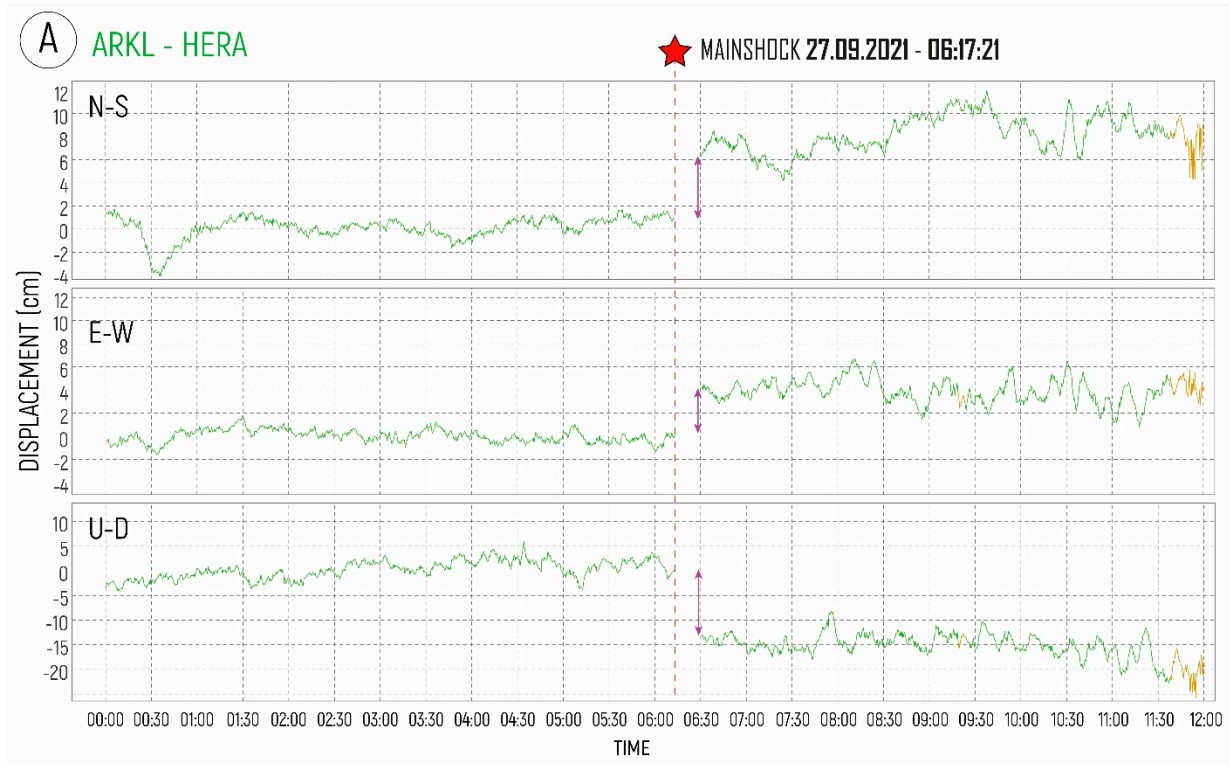

**Figure 6.** *Cont.*

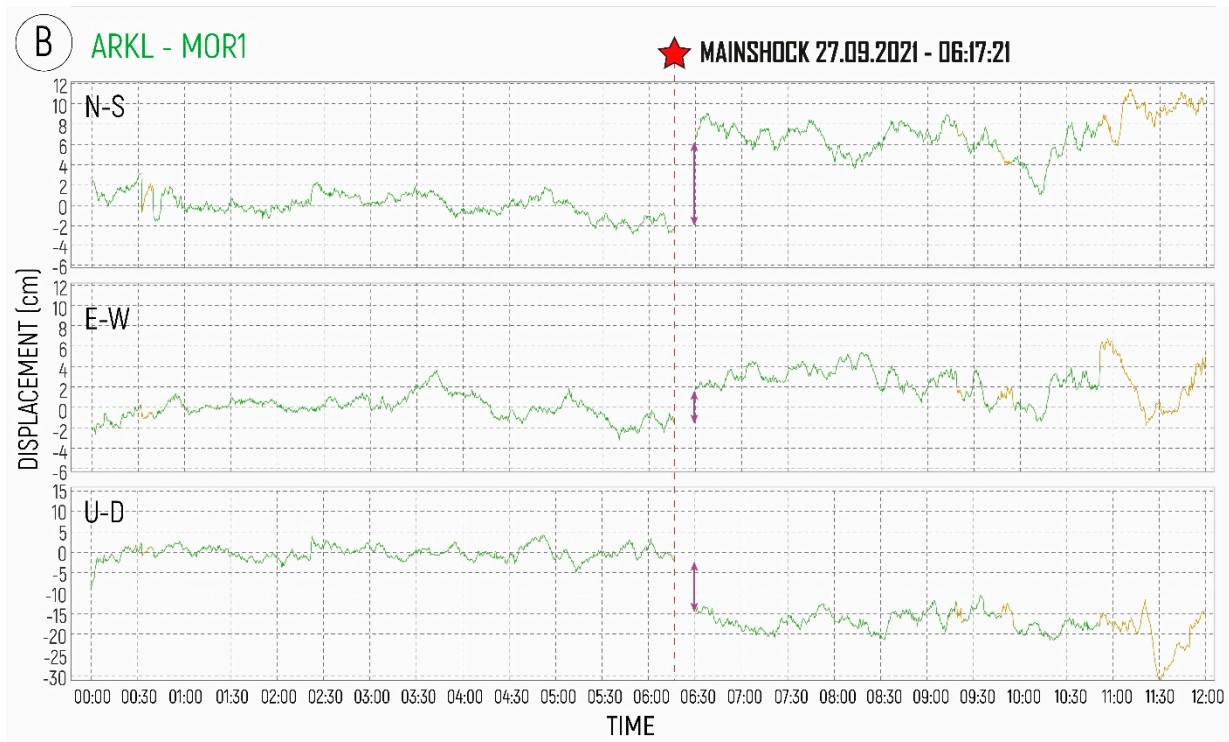

**Figure 6.** Time series during the day of the $M_L$ 5.8 event showing the co-seismic GNSS displacements in the three components of the ARKL station with respect to the HERA base (**A**) and with respect to the MOI1 base (**B**).

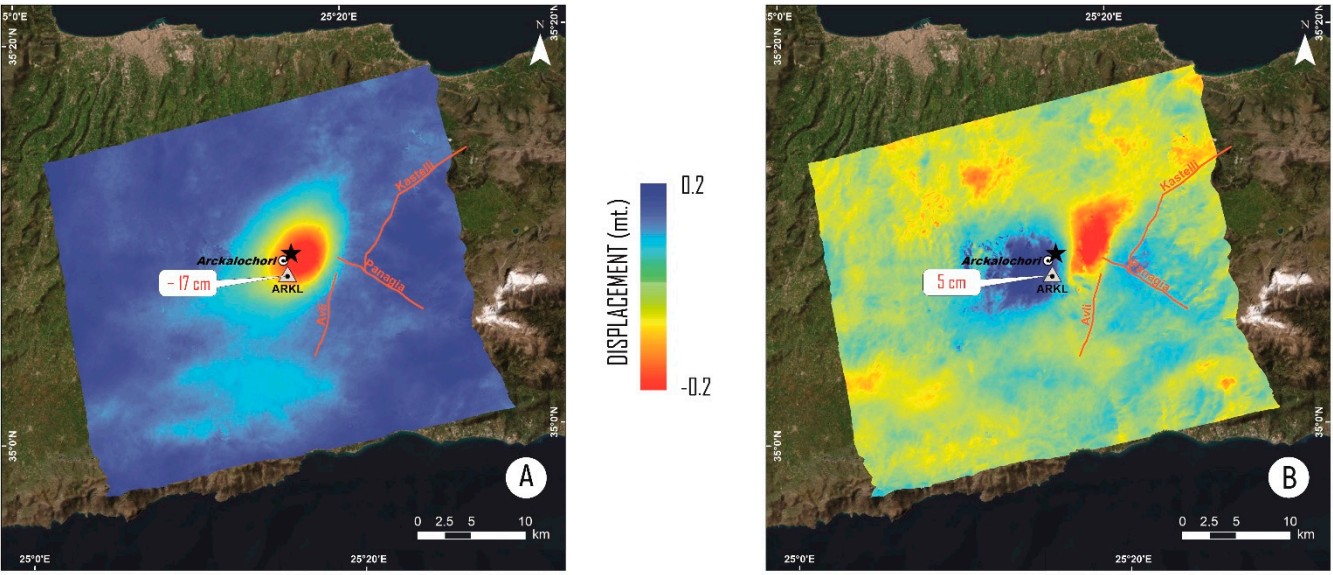

**Figure 7.** Vertical (**A**) and E–W (**B**) InSAR deformation components with relative GNSS ARKL Station co-seismic offset values.

### 3.3. Source Modelling Results

We jointly modelled the dataset of points sampled from InSAR data and displacements obtained from GNSS, assuming that the dislocation occurred over a single surface simplified as a planar geometry, for which all the parameters were left free in the non-linear inversion. The importance of different datasets in modeling was handled by weighting them according to the automatic approach described in [57].

The results of the modelling show a best-fit source with almost purely dip-slip normal kinematics (rake −106°) characterised by a mean slip of about 0.9 m. The reconstructed fault plane dips N–W (strike 214°) with an angle of 50°. The length and width of the uniform slip source are 5.5 and 5.8 km, respectively (the results of analysis from non-linear inversion are added in the auxiliary materials).

**Table 3.** Co-seismic Offset of GNSS PPK processing.

| Stations | Time | Latitude | Longitude | Height |
|---|---|---|---|---|
| ARKL-HERA | Pre Mainschok | 35.1339798770 | 25.2689468500 | 472.48980 |
| | Post Mainschok | 35.1339806210 | 25.2689472780 | 472.33980 |
| | Δ (cm) | 8.25 | 3.90 | −15 |
| | Δ (cm) projected on LOS (Asc): −17.81 | | | |
| | Δ (cm) projected on LOS (Desc): −12.80 | | | |
| ARKL-MOI1 | Pre Mainschok | 35.1339786970 | 25.2689412250 | 472.59870 |
| | Post Mainschok | 35.1339793550 | 25.2689417170 | 472.41760 |
| | Δ (cm) | 7.30 | 4.48 | −18.11 |
| | Δ (cm) projected on LOS (Asc): −20.82 | | | |
| | Δ (cm) projected on LOS (Desc): −12.06 | | | |

This uniform slip model was then extended to 15 km × 13 km and subdivided into elements of 1 km × 1 km to get the slip distribution. The results show a single slip peak distribution that reaches the highest value (~1.0 m) at a depth of ~6 km, with the most dislocation included between about 3 and 12 km of depth (Figure 8).

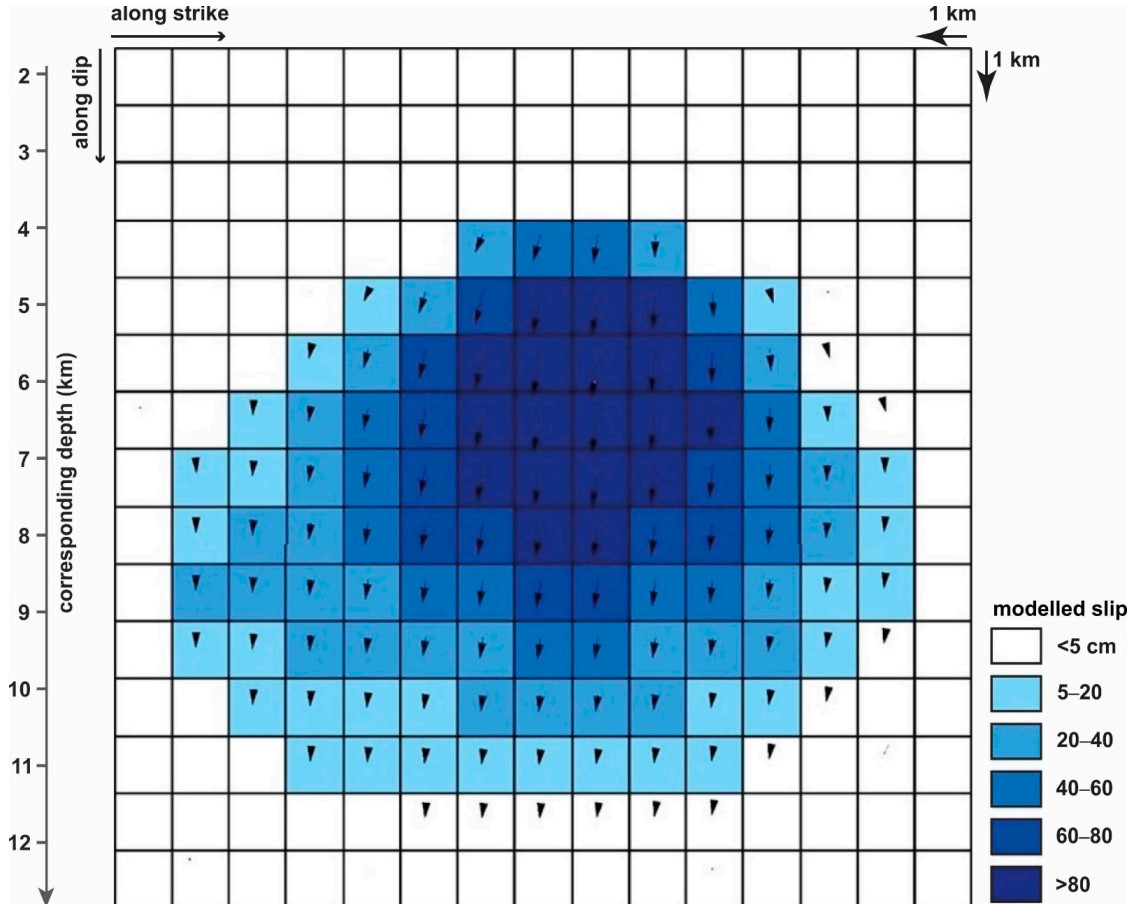

**Figure 8.** $M_L$ 5.8 main event slip distribution. The top of the modelled area is placed at a depth of 1.6 km. Arrows indicate the slip direction at each cell.

The comparison between observed and predicted surface displacement based on the Okada modelling, together with the residuals, basically confirm the high reliability of the obtained solution (Figure 9 and Table 4). The observed, modelled and residuals signals, derived from joint linear least-square inversions for both observations, are shown in Figure 9.

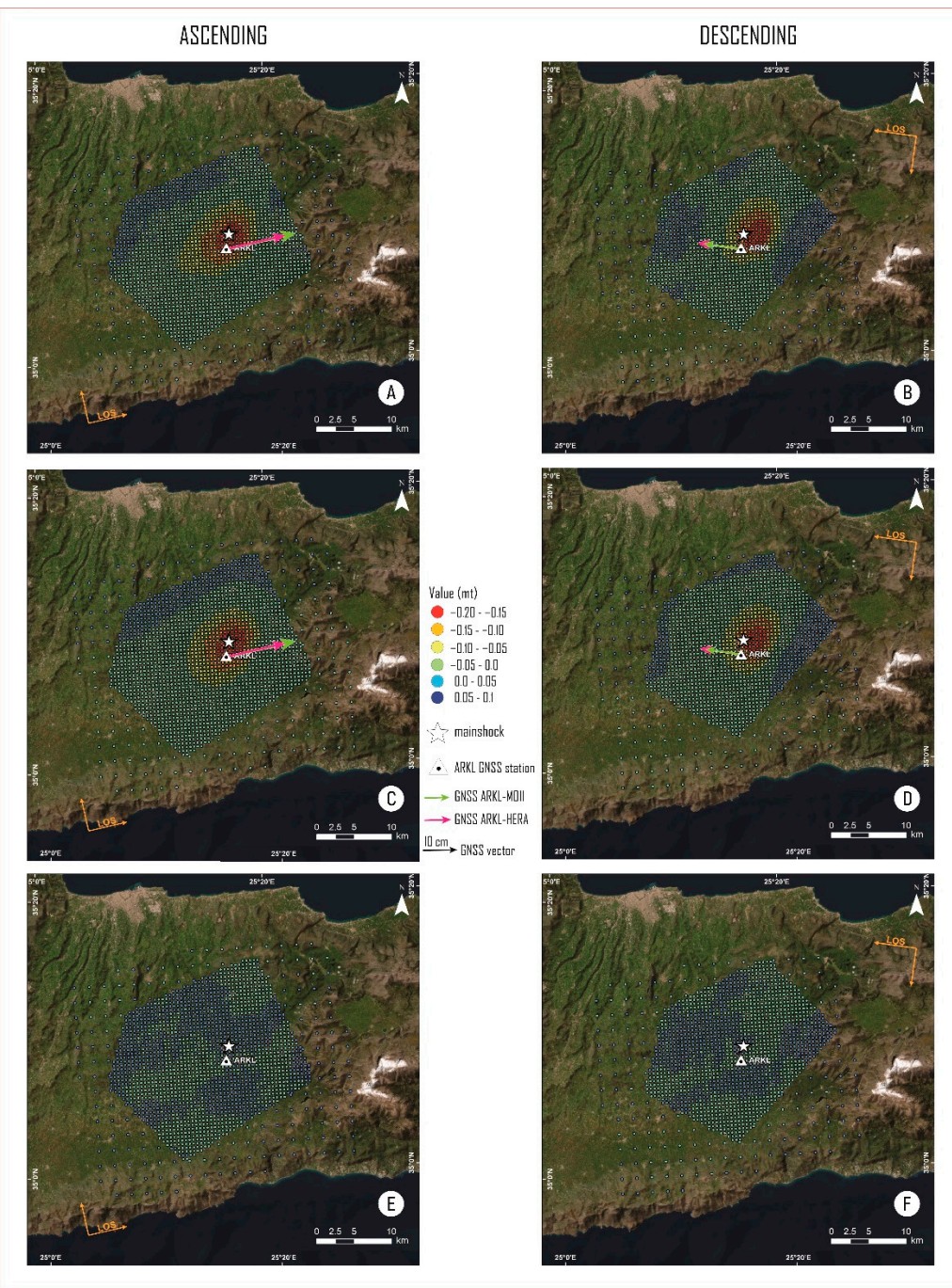

**Figure 9.** Observed (**A,B**), modelled (**C,D**) and residual (**E,F**) maps for displacement obtained from both pairs. The colored arrows represent the displacement vectors of the GNSS, observed and modeled, along the Line Of Sight (LOS) of the satellite. The white star shows the location of mainshock. The white triangle indicates the position of the ARKL GNSS station.

**Table 4.** Observed and modelled co-seismic offset (cm) of GNSS processing projected on LOS direction.

| Stations (Rover-Base) | Ascending Observed | Ascending Modeled | Descending Observed | Descending Modeled |
|---|---|---|---|---|
| ARKL-HERA | −17.81 | −17.56 | −12.80 | −11.68 |
| ARKL-MOI1 | −20.82 | −20.03 | −12.06 | −10.36 |

The results of GNSS data inversion are also shown in Table 4. During the processing, we tried to smooth and reduce the residual patterns for all datasets.

## 4. Conclusions

In this study, a moderate earthquake with the magnitude of $M_L$ 5.8 that struck the island of Crete on 27 September 2021 was analysed. The event occurred in a tectonically active area characterised by multidirectional crustal extension associated to several causes, such as the rapid roll-back of the Hellenic subduction slab, the Aegean mantle wedging and the consequent uplift, the post-orogenic collapse and the arc-parallel stretching.

InSAR images and GNSS data were used to determine the characteristics of the mainshock causative fault and to better understand what happened during the earthquake. The seismic event caused severe damages in the epicentral area and was followed by several aftershocks.

Based on the ascending and descending orbital geometries, the obtained interferograms show the same number of deformation fringes, which indicates that this seismic sequence produced ground movement mainly in the vertical direction (U–D) and only slightly in the horizontal direction (E–W).

The unwrapping phase interferograms confirm coseismic shifts in the mainshock epicentral area. With the aid of two transects traced in the N–S and E–W directions it was possible to emphasise the occurrence of negative displacements along LOS (away from the sensor) of ~18 cm for the ascending orbit and ~20 cm for the descending one. These results are in good agreement with [13,15,58] though some minor differences in the numerical values exist.

The data of three permanent GNSS stations were processed in PPK (Post Processing Kinematic) mode. The station called ARKL positioned in the epicentral area was used as Rover; instead, the HERA and MOI1 stations, a few tens of km from the epicentre, were used as Bases. The results show negative coseismic displacements of about 15–18 cm on the vertical, of about 4–5 cm in the east direction and about 7–8 cm towards the north. Projecting the results to LOS directions, our InSAR results showed good agreement also with the GNSS-based results.

Compared to previously published solutions [58], which highlighted a geodetic seismic moment calculated from the Okada's formalism of $1.14 \times 10^{18}$ Nm (Mw6.0), and a maximum slip of 1.03 m at depths from 3.5 km to 5 km, our model, which is based on a joint inversion of InSAR and GNSS data, seems to provide a better data fit.

In the present research, we also calculated the slip distribution of the source using an algorithm for joint linear inversion of the datasets. The results indicate a major fault striking 214°, dipping 50° towards NW and with an almost dip-slip kinematics (rake: −106). The modelled surface rupture extends at depth up to ca. 12 km and it is partially located in correspondence of the south-southwestern sector of the Kastelli active fault [59], which was clearly considered as a potential seismogenic source in GreDaSS [12].

Moreover, the obtained results indicate a maximum slip of ~1.3 m, occurred at a fault depth of 6 km, comparable to the displacement values suggested by [11] for the same fault (Figures 8 and 10).

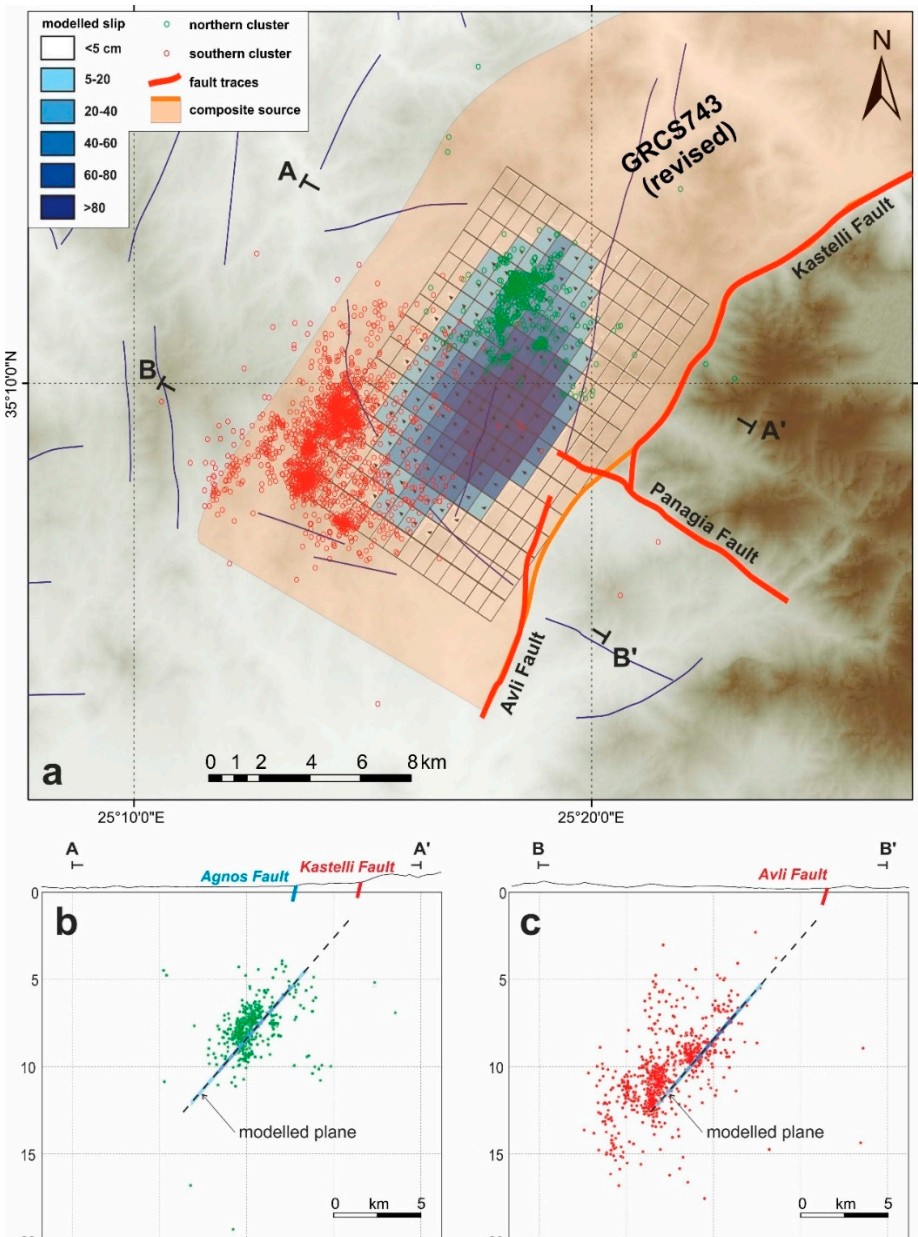

**Figure 10.** (**a**) Map view of the investigated area showing the seismicity from (Reprinted/adapted with permission from Ref. [13]) and the slip modelled fault; (**b**) and (**c**) profiles orthogonal to the modelled slip surface showing the aftershock sequence plotted within a distance of ~3 and 4 km for the A–A′ and B–B′ profiles, respectively. Red and green dots indicate the two seismic clusters. For slip distribution on the modelled surface, see Figure 8. The orange shaded area in (**a**) indicates the proposed revised geometry for the composite seismogenic source GRCS743 [12].

As concerns the modelled source proposed by [15], although their slip distribution shows a single peak similar to ours, some differences relative to the new results should be emphasised: (i) our reconstructed rupture plane is slightly broader, being the maximum dimensions ca. 15 km × 13 km (W × L); (ii) conversely, the maximum slip value is bigger (1.3 m instead of 1.2 m), though occurring at a comparable depth of 5–6 km; (iii) the dip-angle obtained in the present research is slightly smaller (50° instead of 55°); (iv) our best-fit strike is 214° instead of 195°; (v) the minimum and maximum depths are both deeper (3 and ca. 12 km instead of 1.2 and 10 km). After all, these differences possibly explain the better fit we

obtain between the modelled rupture plane (Figure 10) and the hypocentral distribution of the events belonging to the aftershock sequence (e.g., compare with Figure 9 of [15]).

A major outcome of our modelling is the sharp indication that the seismogenic source is represented by the Kastelli Fault (Figure 10) as far as the upward projection of the modelled rupture plane directly and clearly points to the surface in correspondence with the trace of this major tectonic structure (Figure 10b). Accordingly, the hypothesis of a reactivation of the Agnos Fault during the 2021 Crete earthquake, as proposed by some authors [13,15] could be definitely neglected because it would imply an unnatural bending geometry.

Our results are also in agreement with the moment tensor solutions for the main event obtained by several reporting agencies (https://www.emsc-csem.org/Earthquake/index_tensors.php (accessed on 23 September 2022)) suggesting the activation of a normal fault with a mean NW dip-direction and approximately 54° dip-angle.

Similarly, the total seismic moment release of $1.17 \times 10^{18}$ Nm that we estimated, corresponding to a magnitude close to 6, falls within the range of the values reported by the USGS and GCMT, while also the focal mechanism for the modelled source is in agreement with the USGS and GCMT solutions. It should be noted that the aftershocks distribution of the 2021 sequence is clustered in two distinct subvolumes (Figure 10a) suggesting the occurrence of two segments at depth behaving somehow independently. In between, there is a sort of 'silent' volume that is exactly aligned with the westward extension of the Panagia Fault (Figure 10), which is also referred to as Nipitidos Fault by [13].

It is also noteworthy that the latter tectonic structure does not affect at all the Quaternary deposits of the Messara Basin (Figure 1b). This (lack of) evidence, in turn, strongly supports the hypothesis that the Panagia Fault has not been recently reactivated, at the least along its western segment buried under the Quaternary deposits of the Messara Basin. Accordingly, from a seismotectonic point of view this tectonic structure should be considered as an inherited crustal weakness zone crossing some of the NNE-SSW trending active faults mapped in the area (like the Kastelli and Avli faults).

Although inactive, the Panagia Fault had an important, though indirect, role during the 2021 Central Crete event. This role was indeed played either at depth, by partitioning the behaviour of the seismogenic volume (as depicted by the two aftershocks clusters), but probably also within the shallowest crustal volume where the slip surface of the cumulative neotectonic fault (i.e., composite seismogenic source) likely branches at few km depth, say at circa 3–4 km, in correspondence of the intersection with (and a consequence of) the Panagia Fault (Figure 11).

Indeed, in case of a stronger event, the upwards coseismic rupture would have reasonably reached the topographic surface, i.e., linear morphogenic earthquake [60], as largely predicted by empirical relationships [61], therefore cumulating further throw along the Kastelli and the Avli fault scarps, north and south of the Panagia Fault, respectively. This process could have occurred several times during the latest Quaternary, post-LGM [11].

Relative to the composite seismogenic source labelled GRCS743 included in GreDaSS [12], it is worth to emphasise that the cross-cutting relationships with the Panagia Fault and the overstep geometry with the Avli Fault were likely assumed as hard segment boundaries.

From a seismotectonic point of view, the results and interpretations presented and discussed in this paper provide some major lessons.

Firstly, it is the important role that inherited faults (like the Panagia Fault) could possibly play in seismogenesis by altering the stress field close to active faults (like the Kastelli and Avli faults). Secondly, an overstep of a couple of km observed at the surface between fault traces, does not necessarily imply two distinct seismogenic sources as far as fault segments could merge at a few km depth, thus forming a continuous surface. Finally, the seismic hazard prediction implicitly provided in GreDaSS specifically for Crete Island [12] was partially successful in predicting the reactivation of the Kastelli composite seismogenic source (GRCS743) and partially wrong by omitting the Avli segment and missing its contribution.

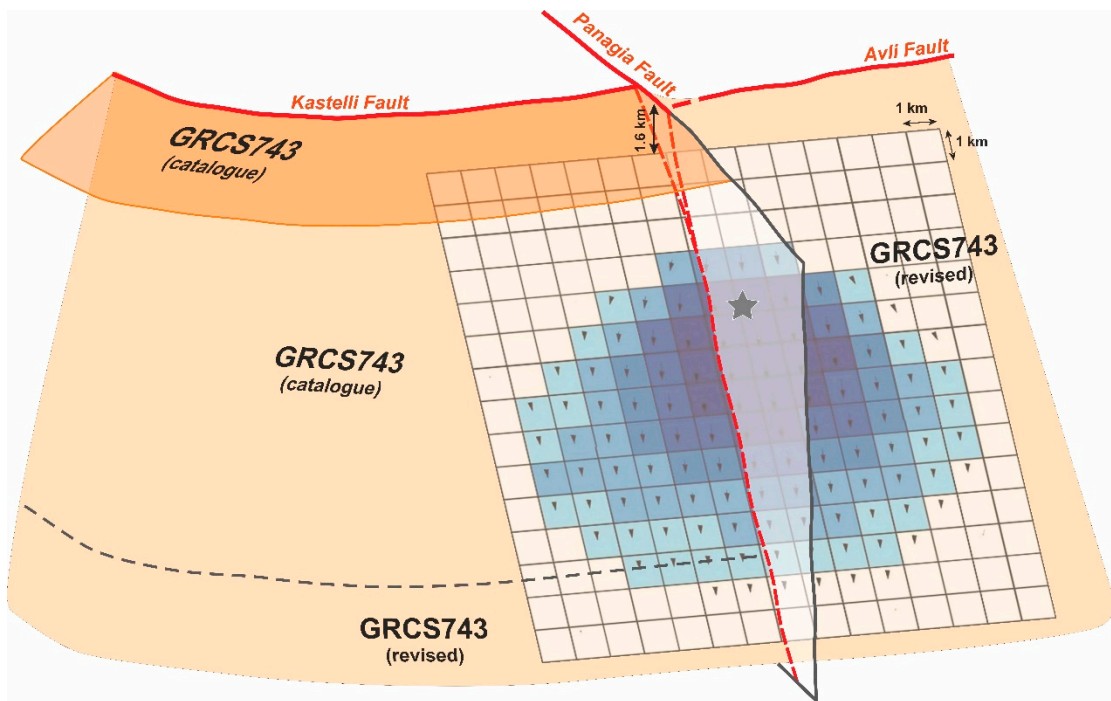

**Figure 11.** Perspective view of the modelled slip surface and its relations with the major faults affecting the investigated volume. The coseismic rupture reactivated the southern sector of the Kastelli seismogenic source, GRCS743 in GreDaSS (Reprinted/adapted with permission from Ref. [12]), and propagated southwards into the Avli Fault and slightly downwards relative to what predicted in the database. The figure also shows the important, though indirect, role of the Panagia Fault during the 2021 seismic sequence. See text discussion. Black star indicate the hypocentre and arrows the slip vector in each modelled cell.

Altogether, the above learned lessons will also contribute to improve the database in Crete, but also in other similar geological and tectonic settings within the Aegean Region.

**Author Contributions:** Conceptualization: N.A.F.; methodology: N.A.F. and Z.G.; SAR software: N.A.F. and Z.G.; GNSS software: N.A.F. and G.C.; validation: All; resources: All; data curation: N.A.F., R.C. and Z.G.; writing—original draft preparation: All; writing—review and editing: N.A.F., Z.G., S.A. and R.C.; supervision: N.A.F., Z.G., A.V. and R.C.; project administration: N.A.F. and A.V.; funding acquisition: N.A.F. and A.V. All authors participated in the writing of the manuscript. All authors have read and agreed to the published version of the manuscript.

**Funding:** This research received no external funding.

**Data Availability Statement:** Sentinel data were made available by ESA in the Copernicus project through the Open Access Hub portal (https://scihub.copernicus.eu/dhus/#/home (accessed on 23 September 2022)); M > 5 historical earthquakes from 1900 up to today from NOA Earthquake Catalogues (https://www.gein.noa.gr/en/services-products/earthquake-catalogs/ (accessed on 23 September 2022)); the composite seismogenic sources included from GreDaSS [12] (http://gredass.unife.it/ (accessed on 23 September 2022)); moment tensor solutions for the main event obtained by several reporting agencies (https://www.emsc-csem.org/Earthquake/index_tensors.php (accessed on 23 September 2022)).

**Acknowledgments:** GNSS data were provided by METRICA SA Company; ENVI® SARscape® (Sarmap, CH) and SNAP (ESA) software were used for InSAR products and source modeling. We are thankful to the three anonymous reviewers and to the Academic Editor for their constructive comments that helped to improve the initial submission of the paper.

**Conflicts of Interest:** The authors declare no conflict of interest.

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
