# Peer review of "The 2021 Greece Central Crete ML 5.8 Earthquake: An Example of Coalescent Fault Segments Reconstructed from InSAR and GNSS Data"

_remotesensing, doi:10.3390/rs14225783_

Round 1

Reviewer 1 Report

Major comments:

1.       The introduction lacks information about previous geodetic studies of the 27 September 2021 central Crete earthquake.

2.       The presented location and name of the ruptured fault is differ from previous studies (Vassilakis et al., 2022; Ganas et al., 2022). This is important because the Kastelli Fault (that is trace can be mapped next to the village of Kastelli) is the fault that was ruptured during the earthquake according to Vassilakis et al., 2022 and Ganas et al., 2022. This fault is named Agnos Fault in Figure 1c and it only shown in Figures 1c and 2 in the Introduction and completely ignored elsewhere. In fact, the Kastelli Fault in this manuscript is farther east and named Geraki Fault in Ganas et al., 2022. This issue is completely ignored in the manuscript.

3.       The confusion in fault name, location and geometry and not taking into account the “other” Kastelli Fault (Agnos Fault here) affect the entire manuscript including the presented model, discussion and conclusions.

4.       It is not clear what software was used to process the InSAR data.

5.       The presented model (Figure 8 and the relevant text) seem to be too smooth and the deepest parts of the displaced area are not shown in the results. The authors should test the smoothing factor (probably decrease smoothing) and present the entire fault rupture distribution.

Minor comments:

Line 47: delete important.

Line 48: change within to near/ north of

Line 59: Strat new paragraph.

Line 72: it is not clear to a potential reader what GreDaSS is.

Line 77: What is the uncertainty of the recurrence interval? Change about 812 years to 812 +- … 800 years.

Section 2.2: What is the typical uncertainties in locations and displacements in the presented RTK software?

Lines 367 and 373: change surface to plane or area. Since we are dealing with blind fault than surface can be confusing, suggesting that the Earth’s surface was ruptured.

Figure 1: Agnos Fault is missing in Figure 1b.

Figure 5: The presented figure 5 is not clear. Probably the resolution is not good enough.

Figure 6: The same as figure 5.

Figure 8: The deepest part of the ruptured area are not shown in the figure. The figure should be changed.

Reviewer 2 Report

This paper provides a very accurate analysis of ground displacement data observed in correspondence of a moderate size earthquake sequence occurred in Crete (Greece) during 2021. Two types of data (GNSS and InSAR) are used and then compared, providing very reliable results on the deformation pattern and their source. The methods of analysis and the data processing techniques are clearly presented. The inversion method is very robust, allowing to constrain properly the source geometry and the associated slip distribution. I have really appreciated the critical discussion of results and the seismotectonic implications from the derived model, which can be extended to many other seismic areas around the world. The paper is of high quality, and it is generally clearly written and illustrated.

The discussion of results is well presented, and the conclusions look consistent with the data analysis. The references are appropriate. I strongly recommend this paper can be published in Remote Sensing.

I have listed below some minor shortcomings or printing errors:

Fig.1: Resolution of this Figure should be improved

At line 307 please modify as:  1 km x 1 km

At line 353 please modify as: 1018 Nm

At line 368 please modify as: 15 km x 13 km

At line 387 please modify as: 1018 Nm

Reviewer 3 Report

Thanks to the authors for their detailed study.

My comments about the study are given in the attached PDF file.

Round 2

Reviewer 1 Report

The submitted manuscript is obviously an improved version of the paper. However, my previous #2 and #3 major comments where not fully addressed in the revised version of the manuscript.

1. The authors do not discuss the fact that previous studies suggested alternative fault model (and geometry).  The authors should explain this point.

2. Since we are dealing with a blind fault then It is not clear from the presented model why the authors decided that the agnos F. was not rupture or why the Kastelli F. was ruptured during the earthquake. Please explain.
